# Empowering a qudit-based quantum processor by traversing the dual bosonic ladder

Long B. Nguyen [1,2,5] ✉, Noah Goss [1,2,5] ✉, Karthik Siva[1,4], Yosep Kim [3], Ed Younis[2], Bingcheng Qing [1], Akel Hashim [1,2], David I. Santiago [1,2] & Irfan Siddiqi[1,2]

High-dimensional quantum information processing has emerged as a promising avenue to transcend hardware limitations and advance the frontiers of quantum technologies. Harnessing the untapped potential of the so-called qudits necessitates the development of quantum protocols beyond the established qubit methodologies. Here, we present a robust, hardware-efficient, and scalable approach for operating multidimensional solid-state systems using Raman-assisted two-photon interactions. We then utilize them to construct extensible multi-qubit operations, realize highly entangled multidimensional states including atomic squeezed states and Schrödinger cat states, and implement programmable entanglement distribution along a qudit array. Our work illuminates the quantum electrodynamics of strongly driven multi-qudit systems and provides the experimental foundation for the future development of high-dimensional quantum applications such as quantum sensing and fault-tolerant quantum computing.

Superconducting circuits have gained recognition as a promising platform in the pursuit of quantum simulation and quantum information processing. Yet, scaling up existing architectures poses formidable challenges in terms of fabrication imperfection, constrained connectivity, and demanding instrumentation overhead. One key strategy to overcome the hardware limitation is to promote a qubit to a qudit and utilize higher energy levels already present in the system to encode or transmit quantum information[1]. In particular, it has been proposed that using qudits may simplify quantum circuit complexity[1,2], allow the native quantum simulation of certain natural systems[3], increase the information capacity and noise resilience in quantum communication[4], and lead to more efficient logical qubit encoding[5–7].

In parallel with the developments of qudits using photons[8,9], cold atoms[10], and trapped ions[11], recent works have highlighted the potential of superconducting circuits for high-dimensional quantum computing[12–16]. However, the progress in superconducting qudits is

constrained by a limited interaction toolkit, with the notable absence of scalable XY-type interactions. This impedes the utilization of superconducting qudits in advancing high-dimensional quantum computing.

In this work, we report an extensible approach to operate high-dimensional solid-state systems based on microwave-induced double-excitation hopping, and then showcase its utility by implementing high-fidelity multi-qubit gates, creating a suite of high-dimensional entangled states, and transferring entanglement across a qudit array. The protocol is platform-agnostic and is compatible with other systems. Notably, our results include the development of a theoretical framework that elucidates the two-photon dynamics in strongly driven coupled-qudit systems, a new method to construct multi-qubit gates entirely based on qutrit interactions, the realization of high-fidelity atomic squeezed states and Schrödinger cat states in qudits, and the synthesis of quantum circuits for high-dimensional operations.

[1]Department of Physics, University of California, Berkeley, CA, USA. [2]Computational Research Division, Lawrence Berkeley National Laboratory, Berkeley, CA, USA. [3]Department of Physics, Korea University, Seoul, Korea. [4]Present address: IBM Quantum, Yorktown Heights, NY, USA. [5]These authors contributed equally: Long B. Nguyen, Noah Goss. ✉e-mail: longbnguyen@berkeley.edu; noahgoss@berkeley.edu

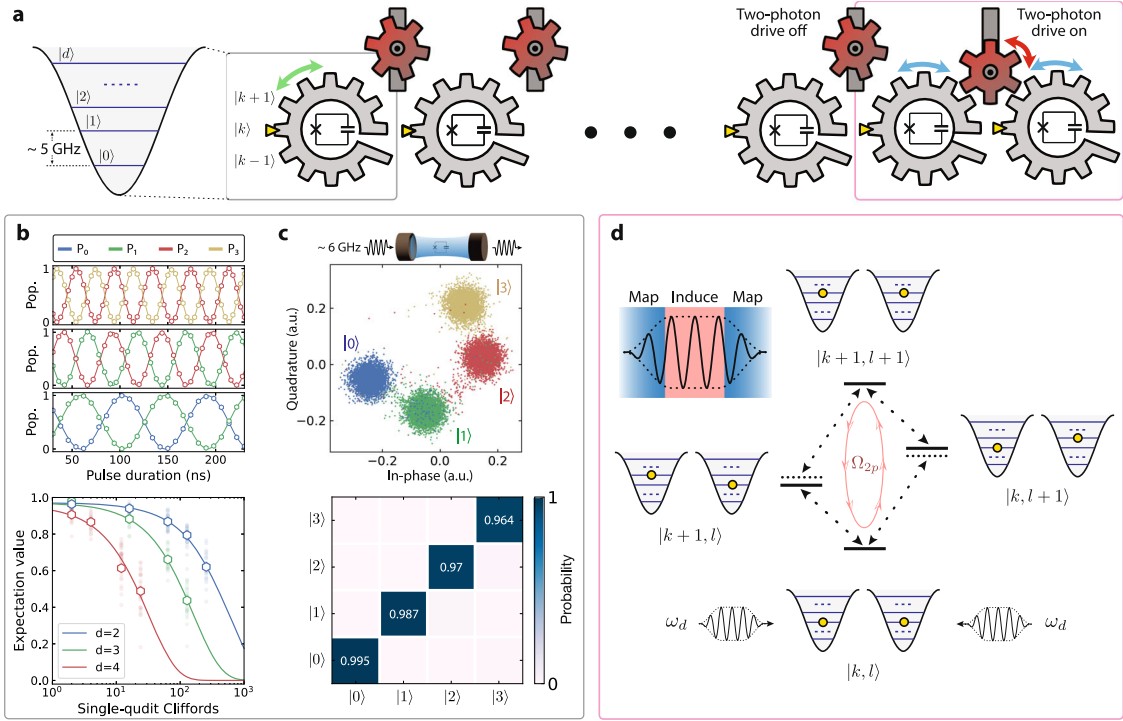

**Fig. 1 | A qudit-based quantum processor. a** Schematic of the high-dimensional system which is constructed by linking individual qudits into an array. Each qudit in the chain is a nonlinear harmonic oscillator comprising of a Josephson junction and a capacitor in parallel, and its eigenstates form a bosonic ladder of $d$ levels within the cosine potential. They are depicted as non-cyclical gears that can be rotated either individually using local controls or simultaneously using two-photon drives. **b** Local control of a qudit. (Top) Rabi oscillations of the populations under resonant microwave drives. The solid lines represent cosine fits. (Bottom) Randomized benchmarking of single-qudit gates. The solid lines represent exponential fits. **c** Readout of a qudit. (Top) A resonator is dispersively coupled to a qudit to measure its state. The probe signal from the resonator distinctly separates into

individual blobs on the IQ plane, corresponding to the qudit being in the states $|0\rangle, |1\rangle, |2\rangle$, and $|3\rangle$. (Bottom) Preparation and measurement confusion matrix of qudit states, which reflects the readout fidelities (Supplementary Note 1). **d** Simplified depiction of the Raman-assisted two-photon-driven dynamics. The coupled-qudits pair form a set of eigenstates, visualized as a dual bosonic ladder. The entangling dynamics takes place in a four-level manifold within this structure. The effect of microwave drives applied to both qudits at a frequency close to the average of the single-excitation frequencies in this subspace is twofold: they first map the relevant qudit states to the dressed frame, then induce a Raman-assisted transition between the bipartite states $|k,l\rangle$ and $|k+1,l+1\rangle$ (inset), resulting in an interaction at a rate $\Omega_{2p}$.

## Results

### Experimental concepts

A superconducting circuit constructed by shunting a Josephson junction with a large capacitor inherently behaves as a nonlinear harmonic oscillator with $d$ eigenlevels forming a bosonic ladder within the potential well[17]. The circuit's eigenstates can be accessed using microwave drives to induce energy excitation, one level at a time. The qudits can thus be naturally visualized using non-cyclical gears, as depicted in Fig. 1a. A high-dimensional quantum device can be built by capacitively connecting the gears into an array, each one serving as a link in the chain with individual control and readout circuitry (Supplementary Note 1).

A snippet of the local control capability is shown in Fig. 1b. On display are the Rabi oscillation data with near-perfect contrast and single-qudit randomized benchmarking results with average native gate fidelities of $\mathcal{F}_g = \{0.99936(3), 0.99909(4), 0.9978(2)\}$ for $d = \{2, 3, 4\}$, respectively. The observed Clifford fidelity decreases for higher $d$ due to the increasing number of native qudit rotations required, with compilations of Clifford groups requiring $\{2, 6, 12\}$ native gates for $d = \{2, 3, 4\}$, respectively (Supplementary Note 1). A dispersively coupled resonator allows the readout of qudit states via heterodyne measurement[18]. As shown in Fig. 1c, the readout signal separates into distinct blobs in the IQ plane for different qudit states, allowing high-fidelity initialization and measurement (Supplementary Note 1).

A natural approach to perform multi-qudit operations on these gears is to engineer their co-rotation which involves the simultaneous

rotation of both qudits. This process embodies a non-energy-conserving two-photon dynamic characteristic of bosonic systems, depicted using the red idler gears in Fig. 1a. We construct the desired interactions for all the high-dimensional subspace manifolds without additional circuitry or instrumentation as follows.

First, a subspace spanned by the bipartite eigenstate $|k,l\rangle$ and its single- and double-excitation with negligible static cross-Kerr can be represented using the dual bosonic ladder as illustrated in Fig. 1d. While transitions involving single-photon hopping are permitted, the direct two-photon hopping process $|k,l\rangle \rightarrow |k+1,l+1\rangle$ is prohibited due to the lack of parity conservation under coherent drives. Notably, within the realm of quantum mechanics, a double excitation of this nature can take place through intermediate states. If these states are solely virtually occupied, the dynamics can be viewed as the multi-partite counterpart of the well-studied Raman process. In multi-qubit systems, this process has been utilized to engineer two-qubit gates in trapped ions with vibrational modes by employing two laser beams at frequencies that combine to match the double-excitation frequency[19]. Here, we harness these dynamics in solid-state systems without the need for ancillary components or additional degrees of freedom.

To simplify the description, we examine the monochromatic driving scenario where two pulses are administered to both qudits at a single frequency $\omega_d$. This frequency is detuned by $\Delta_{1(2)} = \omega_d - \omega_{q,1(2)}$ from those of the relevant single-photon transitions $\omega_{q,1(2)}$, i.e. between states $|k,l\rangle$ and $|k+1,l\rangle(|k,l+1\rangle)$ for qudit 1(2). In this context, the initial effect of the drives involves dressing the qudit states, contingent

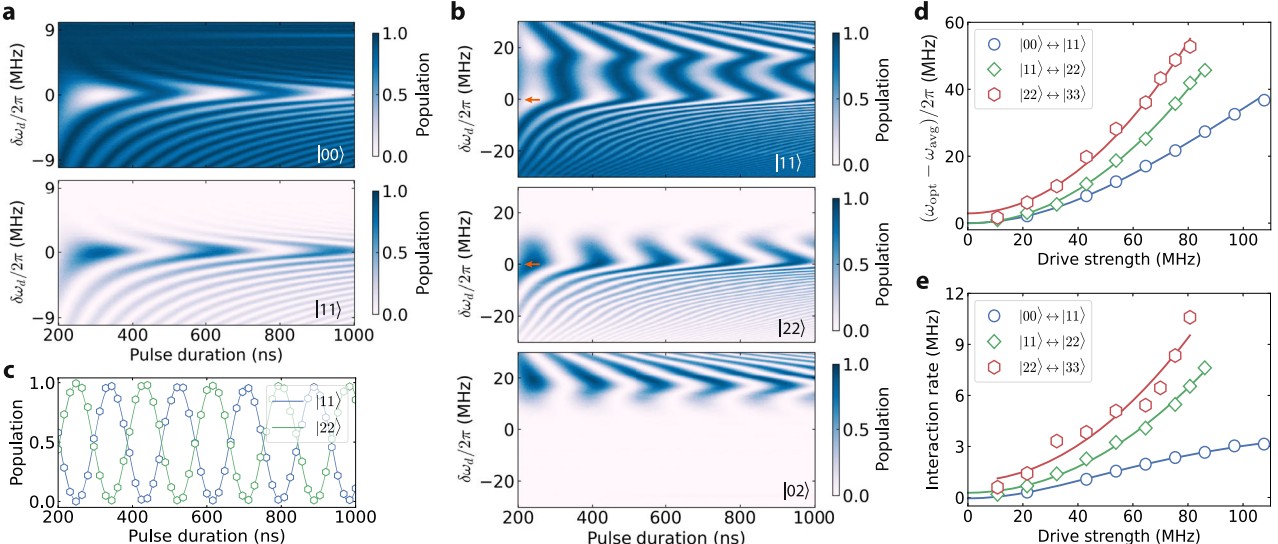

**Fig. 2 | Traversing the dual bosonic ladder. a** Coherent $|00\rangle \leftrightarrow |11\rangle$ population exchange between two coupled qudits. The interaction is induced by monochromatic microwave drives. Here, the drive frequency is shown with respect to the optimal point where the oscillation is the most coherent. The chevron is also quasi-symmetric about this frequency point. **b** Coherent $|11\rangle \leftrightarrow |22\rangle$ and $|11\rangle \leftrightarrow |02\rangle$ population exchange between two coupled qudits. The former interaction is induced by the two-photon Raman process, while the latter originates from the resonant condition between $|11\rangle$ and $|02\rangle$ in the driven frame. **c** Coherent flip-flop oscillation between $|11\rangle$ and $|22\rangle$ states obtained at the optimal drive frequency, indicated by the red arrows in panel **b**. The solid lines represent cosine fits. **d** Optimal driving frequencies at various drive amplitudes. The data are shown with respect to $\omega_{avg} = (\omega_{k,l+1} + \omega_{k+1,l})/2$ in the $y$-axis. The $x$-axis is expressed in terms of the single-photon Rabi rate in the 0-1 subspace of the first qudit. The solid lines are analytical results. **e** Interaction rates of the dual bosonic transitions up to $d = 4$ at various drive amplitudes. All data points are measured at the optimal driving frequencies. Analytical results are shown as solid lines.

upon the fulfillment of the adiabaticity condition[20] (Fig. 1d, inset). With a finite static coupling amplitude $g_{kl}$, the drives subsequently induce coherent two-photon hopping in the interaction frame via sequential single-photon virtual transitions through the hybridized states. Coherent transfer of the population between $|k,l\rangle$ and $|k+1,l+1\rangle$ ensues, with an oscillation rate $\Omega_{2p}$ given as (Supplementary Note 2)

$$\Omega_{2p} = \frac{g_{kl}}{2}\left[(1+\cos\theta_1)(1+\cos\theta_2) + (1-\cos\theta_1)(1-\cos\theta_2)\right], \quad (1)$$

where the angles $\{\theta_i\}$ are defined through the relation $\cos\theta_i = \Delta_i / \sqrt{\Delta_i^2 + \Omega_i^2}$. Here, $\Omega_i$ denotes the driving amplitude on qudit $i$, expressed in terms of the single-photon Rabi oscillation rate between the respective states.

The monochromatic driving scheme is particularly well-suited for the current high-dimensional system, as superconducting qudits inherently possess finite detunings between their energy levels. The drive frequency can be adjusted by varying the ratio between the drive amplitudes, enabling the adjustment of the dressed qudit frequencies (Supplementary Note 2). Employing drives detuned similarly from the pertinent transitions also simplifies pulse calibration to ensure adiabaticity. This approach is readily achievable given the existence of intermediate states that facilitate virtual transitions. The requirements for level detuning are relatively flexible, as a greater detuning permits stronger driving without inducing non-adiabatic effects. Therefore, the presented process is broadly feasible and adaptable across various platforms.

**Traversing the dual bosonic ladder**
We commence the experiment from the bottom of the ladder, starting with the $|00\rangle$ state. As the frequency $\omega_d$ of the microwave drives applied to the qudit pair is swept across the region around the median qudit frequency in the dressed frame, a coherent oscillation between

$|00\rangle$ and $|11\rangle$ appears and forms a chevron pattern with respect to the drive frequency and pulse duration (Fig. 2a). A pulse ramping time of $\tau_r = 100$ ns is used to ensure adiabaticity[20], which is verified by the absence of qudit population in the intermediate states following the drive in all of the following measurements. We note that (i) shorter ramping times can be used, depending on the drive parameters, and (ii) shortcut-to-adiabaticity techniques may help reduce the total interaction times further[20].

The procedure is then extended to induce coherent energy exchange between other bipartite qudit states. Notably, within the $|11\rangle$ and $|22\rangle$ chevron, we observe an energy-conserving exchange interaction between $|11\rangle$ and $|02\rangle$ at certain drive frequencies, as shown in Fig. 2b. This interaction is due to the resonance condition in the dressed frame between these states, previously reported in Ref. 20, further validating our understanding of the strong drive effects in the system. By detecting and avoiding such a condition, we can realize the desired interactions without unwanted effects. As Fig. 2c shows, the fast oscillation between $|11\rangle$ and $|22\rangle$ at the optimal driving frequency (red arrows in Fig. 2b) remains highly coherent for longer than a microsecond without any spurious effect, indicating the robustness of the interaction.

We then repeat the measurement at different drive amplitudes to thoroughly explore the two-photon drive dynamics. We note that microwave line crosstalk and the presence of other energy levels lead to a deviation of the optimal driving frequency from the median frequency between the pertinent multi-qudit levels, which can be accounted for by our analytical model (Fig. 2d). Notably, this reveals that the optimal drive frequency can be modified by changing the ratio of the drives, which can be verified numerically (Supplementary Note 2).

The population oscillation observed at the optimal drive frequency is then used to extract the interaction rate, as shown in Fig. 2e. Our analytical model (Eq. (1)) again exhibits excellent agreement with the experimental data. Notably, the higher we climb up the ladder, the faster oscillations we observe. We attribute this to the larger matrix

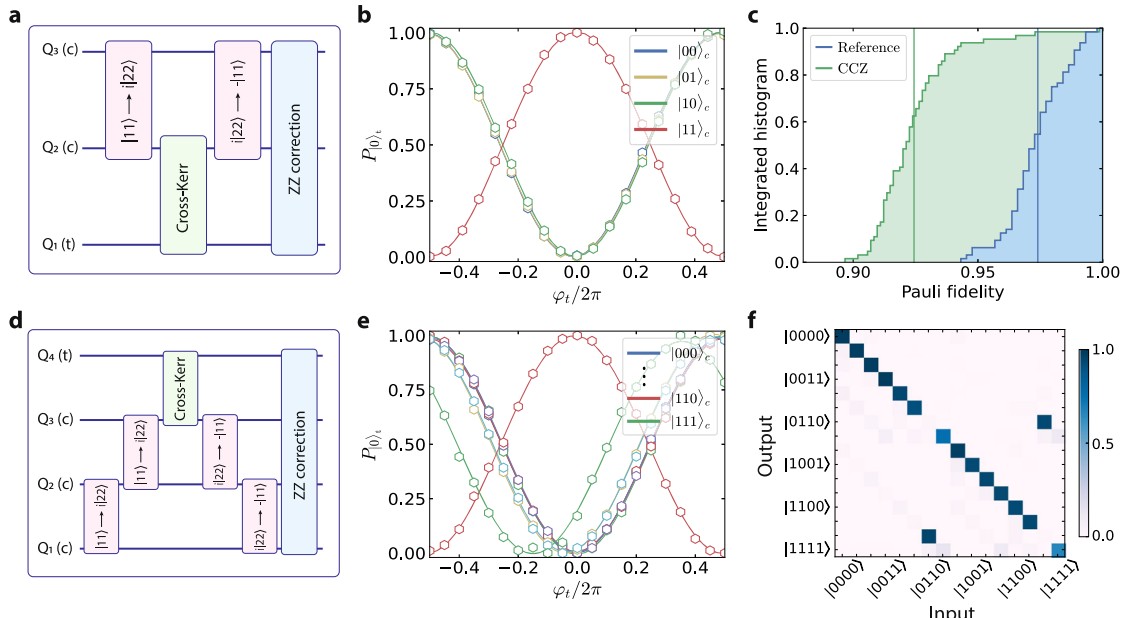

**Fig. 3 | Implementation and verification of multi-qubit gates. a** Gate sequence to implement a CCZ unitary. The qutrit swap gates (pink) are used to shelve and then retrieve the control state $|11\rangle_c$. They sandwich a cross-Kerr gate (green) that induces a Z gate on $Q_1$ if and only if $Q_2$ is in $|2\rangle$. The final stage (blue) is used to correct the residual ZZ phases between the qubits. **b** The three-body operation manifests as the phase shift of the target qubit ($Q_1$) when the control qubits ($Q_2$ and $Q_3$) are in $|11\rangle_c$. The solid lines represent cosine fits. **c** Pauli fidelities of the dressed cycle and the reference cycle from cycle benchmarking. The resulting gate fidelity is $\mathcal{F}_{CCZ} = 96.0(3)\%$. **d** Gate sequence to implement a CCCZ unitary. A cascade of qutrit

swap interactions (pink) is used to shelve and retrieve the respective $|11\rangle_c$ states. In the middle of the sequence is a cross-Kerr gate (green) that induces a Z gate on $Q_4$ if and only if $Q_3$ is in $|2\rangle$. All the residual correlated phases are corrected in the final stage (blue). **e** The four-body operation is effectively revealed through a $\pi$-phase-shift of the target qubit ($Q_4$) for the control state $|110\rangle_c$. The solid lines represent cosine fits. **f** Truth table of the four-qubit Toffoli gate. The target state is shown to be flipped when the control state is $|Q_3Q_2Q_1\rangle = |110\rangle_c$ ($|0110\rangle \leftrightarrow |1110\rangle$). The corresponding truth table fidelity is $\mathcal{F}_{CCCZ} = 92(1)\%$.

elements of the exchange interaction at higher qudit levels. The accelerated interaction crucially allows for shorter gate times that may compensate for the reduced coherence times of these transitions. The small deviation of the obtained data from the analytical solution for $|22\rangle \leftrightarrow |33\rangle$ interaction rates prompts a more detailed theoretical investigation in the future.

We additionally emphasize that while the current discussion focuses on the two-photon dynamics in the $|k,l\rangle$-spanned subspace for equal excitation quanta in each qudit, $k = l$, the process is also applicable for the general case, $k \neq l$, provided there is coupling between the intermediate states $|k+1,l\rangle$ and $|k,l+1\rangle$. For example, it can be leveraged to induce the energy exchange between $|01\rangle$ and $|12\rangle$ in two coupled qutrits (Supplementary Note 3). This feature further substantiates the extensibility of the protocol.

**Synthesis of multi-partite operations**
The higher levels of the qudits can be utilized for shelving, allowing access to otherwise resource-expensive operations[20–22]. Here, we present a new method to synthesize high-fidelity multi-qubit gates using the two-photon interactions. The gate sequence, shown in Fig. 3a, involves sandwiching a cross-Kerr two-qutrit gate[13] on $Q_1$-$Q_2$ between two $|11\rangle \leftrightarrow |22\rangle$ swap gates on $Q_2$-$Q_3$.

First, the swap dynamics is used to substitute and then retrieve the control state $|11\rangle_c$. Then, the implemented cross-Kerr unitary between $Q_1$ and $Q_2$ applies a $\pi$-phase twist to $Q_1$'s $0 - 1$ subspace if and only if $Q_2$ is in its $|2\rangle$ state, effectively leading to a phase shift of the tripartite input state $|111\rangle$. In addition, the off-resonant drives introduce spurious ZZ coupling in the qubit subspace, which we take an additional step to negate using dynamical cross-Kerr pulses[13,20].

The entangling effect can be verified using a modified Ramsey sequence to probe the phase $\varphi_t$ of $Q_1$ for different $Q_2$ and $Q_3$ states[20]. As we apply the CCZ gate sequence, the target phase $\varphi_t$ corresponding

to the control state $|11\rangle_c$ displays a $\pi$ shift relative to the others, signifying the three-body entanglement (Fig. 3b). We then employ cycle benchmarking[23] (CB) to characterize the precision of the realized operation. By comparing the dressed cycle fidelity with the reference cycle fidelity, as shown in Fig. 3c, we estimate a three-qubit gate fidelity of 96.0(3)%.

To further demonstrate the extensibility of this approach, we implement a four-qubit entangling gate by cascading the shelving steps as shown in Fig. 3d. To showcase its versatility, we presently place the cross-Kerr gate between $Q_3$-$Q_4$ and perform the shelve-retrieve operations on the other pairs. Together with the ZZ correction at the end, the sequence effectively applies a $\pi$-phase-shift to the input state $|Q_4Q_3Q_2Q_1\rangle = |1110\rangle$. We again verified this through a conditionality measurement, as shown in Fig. 3e. We note that there exists a coherent error that manifests as a small phase shift in $\varphi_t$ when the control qubits are in $|111\rangle_c$ state.

Since finding the inverse unitary is challenging for a four-qubit circuit, CB cannot be used to determine the gate fidelity efficiently. Alternatively, we validate the operation by sandwiching it between two Hadamard gates on $Q_4$ and measuring the qubit states for different combinations of input states. The truth table summarizing the outcomes in Fig. 3f shows the flipping of the target qubit ($Q_4$) when the control state is $|Q_3Q_2Q_1\rangle = |110\rangle$. The corresponding truth table fidelity is $\mathcal{F}_{CCCZ} = 92(1)\%$, on-par with the best results previously reported in superconducting qubits[22] and trapped-ions[24]. A subsequent cross-entropy benchmarking measurement of the CCCZ gate yields a fidelity of 88(2)% (Supplementary Note 4), implying small phase errors not present in the truth table.

**High-dimensional quantum entanglement**
The capacity of a quantum system is often associated with how efficiently entanglement can be generated within it. The two-photon

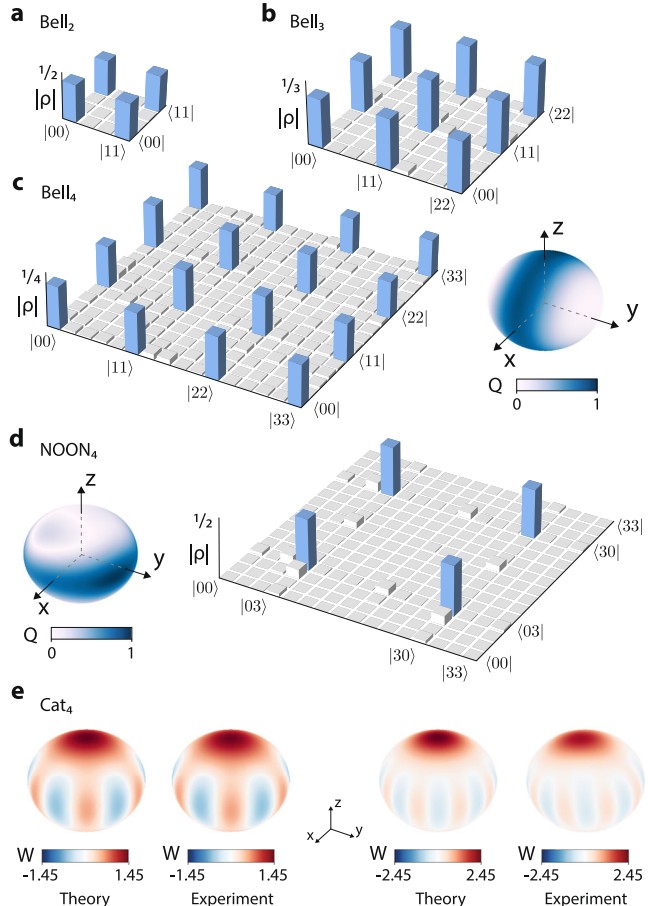

**Fig. 4 | High-dimensional entanglement. a** Density matrix of the qubit ($d = 2$) Bell state with a raw (purified) fidelity of 99.2% (99.9%). **b** Density matrix of the qutrit ($d = 3$) Bell state with $\mathcal{F} = 97.7\%(99.6\%)$. **c** Density matrix (left) and Husimi-Q distribution (right) of the ququart ($d = 4$) Bell state with $\mathcal{F} = 94.3\%(99.3\%)$. **d** Husimi-Q distribution (left) and density matrix (right) of the ququart ($d = 4$) NOON state $|30\rangle + e^{i\alpha}|03\rangle)/\sqrt{2}$ with $\mathcal{F} = 94.6\%(97.3\%)$. **e** Wigner functions of the high-dimensional atomic cat states, $(|00\rangle + e^{i\alpha}|33\rangle)/\sqrt{2}$ with $\mathcal{F} = 98.6\%(99.3\%)$ (left) and $(|000\rangle + e^{i\alpha}|333\rangle)/\sqrt{2}$ with $\mathcal{F} = 80.1\%(90.9\%)$ (right).

interaction is particularly well-suited to propagate entanglement to the higher levels in coupled qudit systems. By traversing the dual bosonic ladder, we proceed to create qudit Bell states with $d = \{2, 3, 4\}$. To verify our findings, we perform quantum state tomography on the output using SU(4) local rotations (Supplementary Note 5). The results are shown in Fig. 4a–c. To estimate the effects of decoherence on the reconstructed states, we compute the fidelity with and without McWeeny purification[25], which finds the nearest idempotent matrix. This purification leads to a significant improvement of the quantum state fidelities compared to the unpurified results, highlighting that decoherence is a primary factor affecting the results. We observe a minor decrease in fidelity for purified states of larger dimension $d$, which is attributable to coherent unitary errors. We provide an estimation of these and tomographic uncertainty due to shot noise in Supplementary Note 6.

Notably, the cascaded two-photon interactions are reminiscent of the two-mode squeezing effect in quantum optics, which is described by the operator $\hat{S}(\zeta) = \exp(\zeta\hat{a}^{\dagger}\hat{b}^{\dagger} - \zeta^{*}\hat{a}\hat{b})$, where $\zeta$ is the squeezing strength, and $\hat{a}$ ($\hat{b}$) is the annihilation operator on the first (second) mode. This results in a resemblance between the high-dimensional Bell states and photonic two-mode squeezed states, $|\Psi\rangle_{2ms} \propto \sum_{N=0}^{\infty} c_N|NN\rangle$, where $c_N$ is a coefficient dependent on $\zeta$ and $N$

is the quanta number[26]. In the present system, $N$ effectively ranges from 0 to $d - 1$.

We verify the squeezing effect by analyzing the Bell$_4$ state using the Husimi-Q quasiprobability distribution (QPD). Here, we use the collective spin coherent state as the basis state with isotropic QPD[27] (Supplementary Note 7). In contrast, a squeezed state would manifest as an elongated strip with a shrunk QPD along a geodesic on the sphere[28]. As shown in Fig. 4c(right), the measured state indeed displays squeezing along the y-axis, confirming the expected outcome. Interestingly, the observed QPD resembles the results associated with atomic squeezed states generated by the two-axis countertwisting Hamiltonian[28].

The NOON state is another important entangled state in quantum metrology, recognized for its exceptional sensitivity in measuring phase differences in interferometric experiments. It is defined as $|\Psi\rangle_{NOON} = (|N0\rangle + e^{i\alpha}|0N\rangle)/\sqrt{2}$, where $N$ equals to $d - 1$ in qudits. We proceed to utilize a combination of a two-photon process and single-qudit rotations to construct a high-fidelity NOON$_4$ state, as depicted in Fig. 4d. Notably, the Husimi-Q QPD verifies that the NOON state exhibits squeezing along the geodesic orthogonal to that of the Bell states.

Atomic Schrödinger cat states, which are superpositions of macroscopically distinct quantum states, belong to another category of entangled states that attract considerable interest[29]. In an $n$-qudit system where each qudit has dimension $d + 1$, the coherent states of interest are $|00\ldots 0\rangle$ and $|dd\ldots d\rangle$[28]. Using the two-photon interaction, we examine two different high-dimensional atomic cat states, $|00\rangle + e^{i\alpha}|33\rangle$ and $|000\rangle + e^{i\alpha}|333\rangle$. To characterize these highly entangled states, we employ the Wigner function, which is known to allow the visualization of cat states due to its inherent capacity to accommodate negative values[30]. The experimental results in Fig. 4e show the fringes characteristic of cat states, verifying our analytical prediction. Importantly, qudit cat states have been proposed as the building block for novel quantum error correction schemes[31], so this experimental implementation may serve as the primitive for future resource-efficient fault-tolerant architectures.

## Distribution of qudit entanglement

The process of swapping populations between the qudit states $|k,l\rangle$ and $|k+1,l+1\rangle$ is functionally equivalent to a subspace energy-conserving swap process $|k+1,l\rangle \leftrightarrow |k,l+1\rangle$. This mechanism is crucial in constructing high-dimensional quantum circuits and is particularly important for distributing entanglement across a qudit array with limited connectivity. To examine this feature, our approach begins with the preparation of a two-qudit Bell$_3$ state (Fig. 5a). Following this, we engage in a two-pronged approach: firstly, we focus on redistributing entanglement among distant qudits, and secondly, we aim to extend entanglement throughout the entire qudit array. This dual strategy explores the breadth of entanglement distribution possibilities in high-dimensional quantum systems with larger Hilbert space compared to architectures based on qubits.

To approximate the effectiveness of the operations, we directly measure the populations of the participating qudit states after implementing the sequences. This focuses on examining the diagonal elements of the multi-qudit density matrices and provides a straightforward approach to assess the impact of the swap operation. The observed transfer of population to adjacent qudits, as depicted in Fig. 5b, c, confirms the successful redistribution of populations across the qudit array.

Additionally, we verify their entanglement by measuring the fidelities of the resulting Bell$_3$ states between Q$_1$-Q$_3$ and Q$_1$-Q$_4$ (Supplementary Note 5). The results show quantum state fidelities of 75.7% for the unpurified density matrices and 96.4% for the purified ones in the case of Q$_1$-Q$_3$, and 53.0% (unpurified) versus 95.7% (purified) for

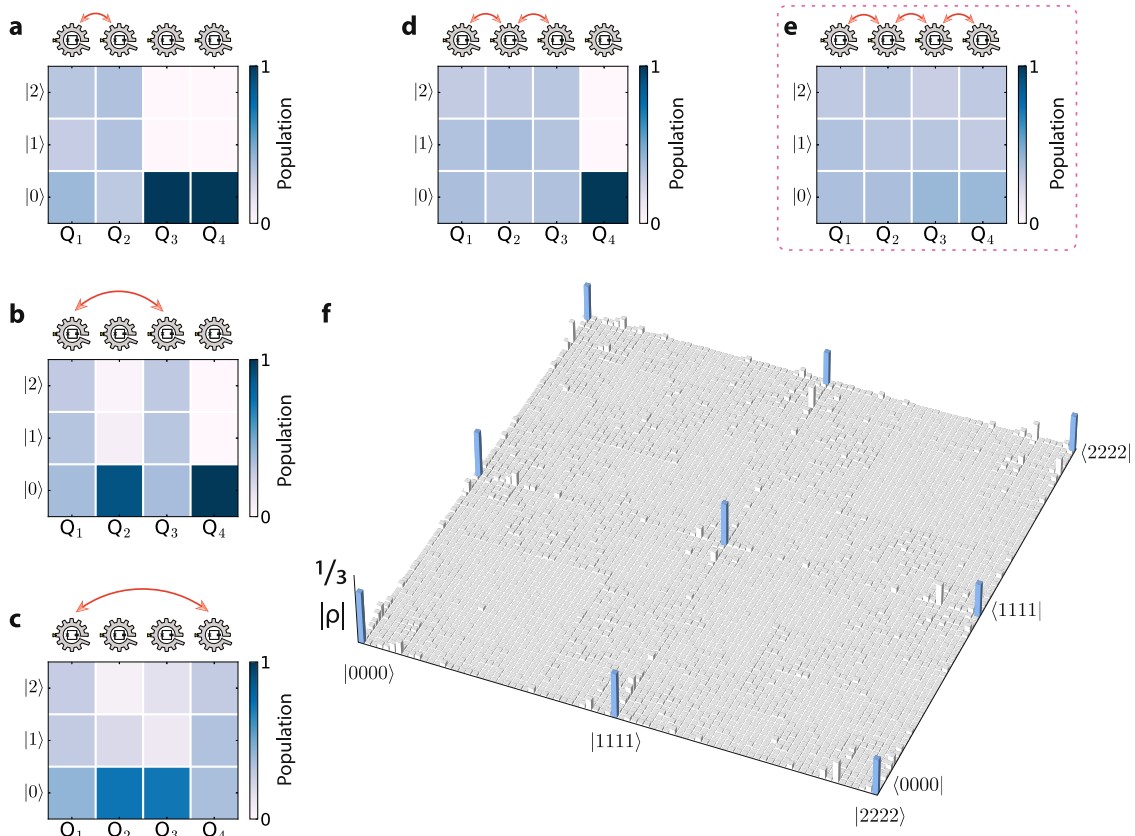

**Fig. 5 | Distribution of qudit entanglement.** Measured populations of (**a**) $Q_1$-$Q_2$ Bell$_3$ state, (**b**) $Q_1$-$Q_3$ Bell$_3$ state, (**c**) $Q_1$-$Q_4$ Bell$_3$ state, (**d**) $Q_1$-$Q_2$-$Q_3$ GHZ$_3$ state, and (**e**) $Q_1$-$Q_2$-$Q_3$-$Q_4$ GHZ$_3$ state. (**f**) Density matrix of the four-qudit GHZ$_3$ state. The raw and purified fidelities are 70.6% and 82.7%, respectively.

$Q_1$-$Q_4$. The substantial discrepancy between the unpurified and purified results indicates that decoherence is the primary source of error in the operations.

Notably, we find that expanding the entanglement space to encompass other qudits requires fewer subspace swaps, in comparison with entanglement redistribution. This leads to the efficient generation of multi-qudit GHZ$_3$ states (Supplementary Note 8). The population measurement results show minimal imperfections, as illustrated in Fig. 5d, e.

We proceed to measure the four-qudit GHZ$_3$ state achieving a fidelity of 70.6% (82.7%) for unpurified (purified) density matrix, which is shown in Fig. 5f. While the difference between these metrics originates from unavoidable decoherence, we attribute the infidelity of the purified result to SPAM errors occurring during the long tomography sequences. In comparison, the three-qudit GHZ$_3$ unpurified (purified) density matrix has a fidelity of 90.8% (96.4%) (see Supplementary Note 5). This implies the need for more efficient and SPAM-free verification methods and benchmarking tools in the future, which we expect to provide a more accurate assessment of the robustness of our procedure.

## Discussion

By combining the ease of quantum control and measurement characteristic of qubit architectures with the concept of an expansive Hilbert space inherent to bosonic systems, we systematically showcase the operational principles of a multi-qudit device based on Raman-assisted two-photon interactions. In contrast with other dynamical processes in strongly driven systems, the reported quantum dynamics remain largely independent of nonlinearity[32,33] or level-matching[34]. In principle, the optimal drive frequency can be tuned in situ by adjusting the drive amplitudes and their relative ratio

(Fig. 2d and Supplementary Note 2). These advantages enhance the extensibility and adaptability of the process, providing a crucial interaction repertoire to high-dimensional solid-state systems. Our findings lay the groundwork for future investigations into superconducting qudits, drawing insights from both bosonic and atomic research advancements.

As the simplicity and robustness of fixed-frequency superconducting circuits with fixed coupling have allowed them to be scaled up to devices integrating hundreds of qubits, our demonstrations using the same type of device without additional circuitry or instrumentation represent a potentially transformative approach to advancing quantum technologies through high-dimensional quantum operations. On the one hand, this work opens the door for exploring quantum sensing and quantum information processing with high-dimensional systems, and on the other hand, the results encourage the future development of bosonic encodings, such as spin-cats[31], using superconducting qudits. Concurrently, the versatility of the protocol motivates its adaptation to implement highly connected quantum architectures. Looking forward, the observed decoherence-limited performance metrics underscore the necessity of developing hardware with enhanced noise protection for higher levels.

## Data availability

All data reported in this work are available at Figshare/25903732.

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

## Acknowledgements

We are grateful to Liang Jiang, Konstantin Nesterov, Seth Merkel, and Ken Xuan Wei for valuable discussions. We thank John Mark Kreikebaum for fabricating the experimental device. This work was supported by the Office of Advanced Scientific Computing Research, Testbeds for Science program, Accelerated Research in Quantum Computing Program, Office of Science of the U.S. Department of Energy under Contract No. DE-AC02-05CH11231. Noah Goss acknowledges funding from the National Science Foundation under Grant No. 2210391. Yosep Kim acknowledges funding from the Creation of the Quantum Information Science R&D Ecosystem through the National Research Foundation of Korea (NRF), grant No. 2022M3H3A106307411.

## Author contributions

L.B.N. conceptualized and organized the experiment. N.G. established the high-dimensional metrology and calibration framework. L.B.N. and N.G. acquired the data and analyzed the results. K.S. developed the theoretical model. Y.K. assisted with visualization and devised the quasiprobability distribution analysis. E.Y. developed the algorithms to optimize the high-dimensional circuits. B.Q. simulated the tomography shot noise, assisted with the Wigner function analysis, and provided the chip image. A.H., D.I.S., and I.S. oversaw the experimental effort.

## Competing interests

The authors declare no competing interests.
