## [Peer Review File · Nature Communications]

Empowering a qudit-based quantum processor by traversing the dual bosonic ladderREVIEWER COMMENTS

Reviewer #1 (Remarks to the Author):

In this manuscript, Nguyen et al. present a series of results about a multi-qudit device. At the core of their results is a novel technique to realize a two-photon gain/loss mechanism between qudits. They demonstrate that they can realize this interaction with high fidelity, and then proceed to demonstrate several ways in which this interaction can be used. In particular, the authors demonstrate the preparation of multi-qudit entangled states.

The results presented in the manuscript are certainly of very high quality: the device itself is state-of-the-art and the fidelities reported for the various operations are very high. This indicates that the authors took great care in optimizing all stages of the experiment. Moreover, to the best of my knowledge, the two-photon drive scheme presented here is novel.

Additionally, as the authors remark, this type of interactions can be realized in "standard" multi-transmon devices, and does not require additional hardware. I believe these results will spur more interest into multi-qudit devices by other groups having access to similar devices. As qudit devices and multi-qudit operations are less studied than their qubit counterpart, I believe that the present manuscript constitutes an important milestone in this developing subfield.

However, in my opinion, the manuscript contains too many examples. As a result, the reading is very dense. In particular, a good portion of the manuscript relies heavily on the supplemental material to explain details.

Nevertheless, for the reasons stated above, I recommend publication of the present manuscript in Nature Communications.

Comments:

I am slightly confused by the sentence: "It is crucial to highlight that the two-photon dynamics here resemble an ancilla-assisted tunneling process, in contrast with other high-order processes that rely on the system's nonlinearity". Looking at Fig 2e which looks like a

quadratic relation between drive strength and coupling rate, I'm tempted to analyze the induced interaction as an induced second-order interaction activated from the four-wave mixing term of the Josephson Junction (JJ). I could agree that it's easier to understand the current scheme as a level-assisted transition, but ultimately all microwave-induced couplings come from higher-order (i.e. ≥ 4) process of the DJs ? In its current form, this sentence seems either confusing or misleading to me.

- In my initial reading of the manuscript, I was confused by the fact that multiqubit states are represented on a single Wigner sphere. The mapping becomes clear when digging into the SM however, but I would suggest adding a sentence about this in the main text.

Reviewer #2 (Remarks to the Author):

In this manuscript, the authors reported a new way to operate superconducting qudits in high-dimensional quantum computing. This operation is based on a Raman-assisted two-photon interaction in superconducting circuits. They demonstrated the capability of this operation via the implementation of multi-qubit gates, creation of various high-dimensional entangled states, and the distribution of entanglement across multiple qudits. In addition to the experimental work, the authors also presented a theoretical framework for understanding the two-photon dynamics in strongly driven coupled-qubit systems. The authors showcased the utility of this framework by realizing atomic squeezed state and Schrodinger cat states in qudits with high fidelity.

I found the presented research was carried out systematically, the manuscript is well written and technically sound. The demonstrated microwave two-photon interaction is an important addition to the toolbox of high-dimensional quantum computing with superconducting qudits. Therefore, I recommend publication of this manuscript after the authors address my comments/questions below.

1. I would like the authors to explain what parameters of the qubit set the adiabaticity condition for the drives of the two-photon hopping process. I understand that this topic was discussed in another recent work by the authors (Ref. 20). But I would really appreciate the authors for an intuitive understanding.

From the data shown in Fig. 2C, it appears that a pi pulse which inverts the population between $|11\rangle$ and $|22\rangle$ state takes about 300 ns. This is relatively slow comparing to the state-of-art in superconducting qubits, where such a pulse for a transmon qubit is typically 100 ns or less (see, for example, Krinner et al., Nature 2022). Therefore, it is desirable to reduce the pulse duration. Furthermore, this appears to be critical for improving the gate fidelity as the authors claimed that decoherence is the major source for gate infidelity.

However, it is not clear to me how much shorter this pulse could be reduced to given that the pulse ramping time is 100 ns which is required by the adiabaticity condition. It will be helpful to if the authors can elaborate on their plan to reduce the gate duration.

2. In Fig. 2E, why the data shows relatively large deviation from the analytical mode for interaction rate between $|22\rangle$ and $|33\rangle$ state? It seems that the data for drive strength between 30 and 50 MHz are following a different scaling comparing to the rest.

3. In the section “Distribution of qudit entanglement”, it is shown that the unpurified quantum state fidelity of Bell-4 state (53.0%) is significantly lower than that of the Bell-3 state (75.7%). I would like the authors to provide a more detailed explanation for this dramatic reduction in state fidelity.

4. The authors attribute the infidelity of the purified quantum states to SPAM errors occurred in the tomography sequence. However, there is no discussion on SPAM errors in these experiments. I would like to ask the authors to provide more details on this topic to support their assessment.

Reviewer #3 (Remarks to the Author):

Nguyen and Goss, et al. present experiments on a device with 4 coupled superconducting transmon circuits. Each of these circuits has an anharmonic energy spectrum with distinct transition frequencies between successive pairs of energy levels and can therefore encode a qudit. They controllably address transitions between states $|k,l\rangle$ and $|k+1,l+1\rangle$ by driving close to the average frequency between the two individual transition frequencies. After

characterizing this operation, they combine it with some of their prior work [references 13,20] to create and measure multi-qubit and multi-qudit entangled states.

I find the manuscript fairly well written and pedagogical although some parts, in particular some of the terminology the authors use, require further clarification. The experiment itself is a true tour-de-force and the data is well analyzed and supported by an extensive supplementary material. While the conceptual novelty of the $|k,l\rangle \leftrightarrow |k+1,l+1\rangle$ transition is rather modest, the authors use it to generate highly entangled qudit states with very good fidelity. I believe that this work could be of interest to the broad readership of Nature Communications, but, before making a recommendation, I would like the authors to address the following points:

A) The authors convincingly outline in the supplementary materials that this approach does not rely on the anharmonic spectrum of a transmon circuit but instead focuses on transitions through virtual states and is therefore more general. However, their experiments are of course performed with transmon circuits and so it is natural to compare to previous experiments in the field. In particular, the monochromatically driven $00 \leftrightarrow 11$ transition in ref 34 seems very similar to the transition shown here. Could the authors please comment in-depth on the similarities and differences between these two transitions?

B) Following up on the previous question: as far as I can see, these gate processes could be framed in terms of a four-wave mixing process enabled by the transmon nonlinearity. In that case the resonance condition would simply be that the two transition frequencies ($k,l \leftrightarrow k+1,l$ and $k,l \leftrightarrow k,l+1$) need to be equal to the sum of two drive frequencies ($\omega_1 + \omega_2$). One could then see the process the authors demonstrate as the special case where $\omega_d = \omega_1 = \omega_2$. Maybe the authors could explain if this is indeed the case? Does their system have capabilities that go beyond this four-wave mixing description in coupled transmons?

C) The authors state that “the optimal drive frequency can be tuned in situ by adjusting the drive amplitudes and their relative ratio.” As far as I can see, the tuning of the optimal drive frequency is not shown experimentally in this work. If such data exists it would be an interesting addition to the SI. Otherwise, the statement should maybe be amended to clarify that it refers to a theoretical prediction.

D) “native gate fidelities of $F_g = \{0.99936(3), 0.99909(4), 0.9978(2)\}$ for $d = \{2, 3, 4\}$,” Are the fidelities extracted from the fit to the data in figure 1b? If so, please mention this and add a reference.

E) Are the lines in Fig1b and 2c fits or a guide for the eye?

F) “(Bottom) Preparation and measurement confusion matrix of qudit states, which reflects the readout fidelities.” Could the authors please give a definition in SI or cite a reference.

G) “While transitions involving single-photon hopping are permitted, the direct two-photon hopping process $|k, l\rangle \rightarrow |k + 1, l + 1\rangle$ is prohibited due to parity.” This could be formulated more clearly. Presumably the authors mean the lack of parity conservation under a coherent drive?

H) Fig2: was $\Delta\omega_d$ defined in the text or figure caption? What is meant by symmetric point? The data looks asymmetric.

I) Fig2 caption and other occurrences throughout the text and supplementary materials: “The y-axis is normalized with respect to ω_{median} ,” In what sense is this a normalization?

J) “median qudit frequency” What is meant by median in this context? Do the authors perhaps mean average?

K) Figure 2d: Why is this frequency shift positive? If ω_{median} is the undriven average qubit frequency and the anharmonicity is negative, then the a.c. Stark shift will generally also be negative. This means that the optimal drive frequency given by the new (smaller) average qubit frequency should be smaller than ω_{median} . Is this positive frequency shift of the resonance condition due to the fact that Q2 is driven in the straddling regime (between the frequencies of the $0 \leftrightarrow 1$ and $1 \leftrightarrow 2$ transitions)? What are the expected frequency shifts of the individual 0-1 transitions of the two qubits under this drive?

L) “which can be accounted for by our analytical model (Fig. 2d).” Please specify which equation is used to get this curve. What are the fit parameters? Is the microwave crosstalk indeed accounted for by that model?

M) Same as above for Fig 2e :” Our analytical model again provides an excellent fit to the experimental data.”

N) “access to otherwise impossible operations” Please explain why these operations would otherwise be impossible.

O) “is FCCZ = 92(1)%, on-par with the best results previously reported in superconducting

qubits [24] and trapped-ions [26].” It would be good to also refer to the lower fidelity estimate given in the SI (88%) by a different method.

P) “for novel fault-tolerance schemes” Incomplete sentence?

Q) All figures that include bar graphs: it seems like only the bars corresponding to the density matrix elements that would be non-zero for an ideal state are coloured blue. This is somewhat visually misleading. It would be better to find a different color scheme and to include wireframe boxes depicting the ideal values

R) SI Figure S5: “(b) Two-photon swap rate Ω_{2p} as a function of the drive amplitude and qudit detuning Δ_{12} . Even when the energy levels $|k + 1, l\rangle$ and $|k, l + 1\rangle$ are far-detuned, it is possible to activate the two-photon swap, albeit requiring more drive power.” For what drive frequency is this simulated?

S) SI: “and compute the energy difference $\sim \omega(i) q$ between the states $|0\rangle$ and $|1\rangle$.” Is this what is meant by exact diagonalization (numerical ED) in Fig. S5? If so, please mention that here.

T) The visual representations on Fig.S1 give a good qualitative overview of the system parameters, but they are not quantitative enough. Could the authors please also add a table including frequencies, anharmonicities, coupling strengths between qubits as well as qubits and resonators, linewidths of resonators. Please also add average T_1, T_2, T_2E .

U) A scale bar is missing in figure S1a

Dear Reviewers:

Thank you for your efforts in reviewing our manuscript "Empowering a qudit-based quantum processor by traversing the dual bosonic ladder". We are glad that Reviewers #1 and #2 found the submitted manuscript suitable for publication, while Reviewer #3 believed that our work could be of interest to the broad readership of Nature Communication. Based on the Reviewers' insightful comments and suggestions, we have modified and improved the manuscript accordingly. A summary of changes is included below, and our responses to the Reviewers are appended. We hope that the Reviewers will find the revised manuscript satisfactory and ready for immediate publication.

Summary of changes

- Figure 1c, 2c, 3b, 3e: we clarified that the solid lines are from fitting.
- Figure 2: we modified the definition of $\delta\omega_d$ and ω_{avg} (previously ω_{median}).
- Experimental concept, page 1: we provided a reference to the Supplementary Material Note 1 regarding the local rotation fidelities.
- Experimental concept, page 2: we further clarified the parity conservation concept.
- Experimental concept, page 3: we discarded the sentence starting with "It is crucial" and appended the extensibility feature of our protocol to the previous paragraph.
- Traversing the dual bosonic ladder, page 4: we further commented on the adiabaticity condition and what can be done in the future to speed up the interaction.
- Traversing the dual bosonic ladder, page 4: we added the text implying how the ratio between the drive amplitudes can be used to change the optimal driving frequency and clarifying how to find the analytical models used for fitting.
- Traversing the dual bosonic ladder, page 4: we commented on the deviation between the experimental data and the analytical model.
- Synthesis of multi-partite operations, page 5: we clarified the detail about synthesis of multi-qubit gates.
- Synthesis of multi-partite operations, page 5: we added the fidelity obtained using XEB.
- High-dimensional quantum entanglement, page 6: we have revised the statement regarding the purified state fidelity.
- High-dimensional quantum entanglement, page 6: we added a sentence to point the reader to the Wigner function reconstruction in Supplementary Materials Note 7.
- High-dimensional quantum entanglement, page 6: we revised the last sentence.
- Discussion, page 7: we added a cross-reference to Fig. 2c and Supplementary Material Note 2 regarding the tunability of the optimal driving frequency.
- Acknowledgments: we added a note to thank John Mark Kreikebaum and an additional funding source.
- Data availability: we added the figshare link to the data set in .txt format.
- Figure S1a: we added a scale bar.
- Supplementary Note 1: we added a Table to summarize the relevant qudit parameters.
- Supplementary Note 1: we added the method used to estimate the local gate fidelities.
- Supplementary Note 1: we added a definition for readout fidelities.
- Supplementary Note 1: we added a reference for fabrication details.

- Figure S5: we revised the definition of the y-axis of panel c.
- Supplementary Note 2: we added a subsection on adiabaticity.
- Supplementary Note 2: we added a comment on how the optimal driving frequency can be tuned by adjusting the drive ratio.
- Supplementary Note 2: we added a clarification on the difference between our scheme and the one reported in Ref. 34.
- Supplementary Note 2: we amended the caption of Fig. S5b and discussion thereof to emphasize the drive ratio and frequency.
- Supplementary Note 2: we amended the explanation of Fig. S5c following Eq. S15 to clarify what each approach presented means.
- Supplementary Note 2: we added a remark to emphasize which models are used to fit the data.
- Supplementary Note 5: we added a discussion regarding the visualization of the state tomography and the availability of the data in matrix form.
- Supplementary Note 6: we added a discussion on tomography errors.
- Supplementary Note 6: we added a discussion on crosstalk and spectator errors.
- Supplementary Note 8: we added a subsection to describe the Bell state redistribution.

Responses to Reviewer #1

Comment 1

The results presented in the manuscript are certainly of very high quality: the device itself is state-of-the-art and the fidelities reported for the various operations are very high. This indicates that the authors took great care in optimizing all stages of the experiment. Moreover, to the best of my knowledge, the two-photon drive scheme presented here is novel. Additionally, as the authors remark, this type of interactions can be realized in "standard" multi-transmon devices, and does not require additional hardware. I believe these results will spur more interest into multi-qudit devices by other groups having access to similar devices. As qudit devices and multi-qudit operations are less studied than their qubit counterpart, I believe that the present manuscript constitutes an important milestone in this developing subfield.

Response:

We thank the Reviewer for the positive remarks. We hope our results will help propel the field of superconducting qudits forward.

Comment 2

However, in my opinion, the manuscript contains too many examples. As a result, the reading is very dense. In particular, a good portion of the manuscript relies heavily on the supplemental material to explain details. Nevertheless, for the reasons stated above, I recommend publication of the present manuscript in Nature Communications.

Response:

We thank the Reviewer for the positive comment and the recommendation for publication. We originally constructed the manuscript to be compatible with journals having a length restriction, and at the same time wanted to demonstrate the significance of the interaction scheme as much as we could. This resulted in a seemingly dense manuscript. We will learn from this experience and prepare manuscripts better in the future.

Comment 3

I am slightly confused by the sentence: "It is crucial to highlight that the two-photon dynamics here resemble an ancilla-assisted tunneling process, in contrast with other high-order processes that rely on the system's nonlinearity". Looking at Fig 2e which looks like a quadratic relation between drive strength and coupling rate, I'm tempted to analyze the induced interaction as an induced second-order interaction activated from the four-wave mixing term of the Josephson Junction (JJ). I could agree that it's easier to understand the current scheme as a level-assisted transition, but ultimately all microwave-induced couplings come from higher-order (i.e. ≥ 4) process of the DJs ? In its current form, this sentence seems either confusing or misleading to me.

Response:

We thank the Reviewer for the insightful comment. We have now discarded this statement to avoid causing confusion.

Comment 4

In my initial reading of the manuscript, I was confused by the fact that multi-qudit states are represented on a single Wigner sphere. The mapping becomes clear when digging into the SM however, but I would suggest adding a sentence about this in the main text.

Response:

We thank the Reviewer for raising this concern. The cited paper (R. P. Rundle et al.) specifically discusses the visualization of multi-qudit entangled states using phase-space representation. We have now added a sentence to guide the reader toward the appropriate section in the Supplementary Material.

Responses to Reviewer #2

Comment 1

I found the presented research was carried out systematically, the manuscript is well written and technically sound. The demonstrated microwave two-photon interaction is an important addition to the toolbox of high-dimensional quantum computing with superconducting qubits. Therefore, I recommend publication of this manuscript after the authors address my comments/questions below.

Response:

We thank the Reviewer for the encouraging assessment and the recommendation for publication.

Comment 2

I would like the authors to explain what parameters of the qubit set the adiabaticity condition for the drives of the two-photon hopping process. I understand that this topic was discussed in another recent work by the authors (Ref. 20). But I would really appreciate the authors for an intuitive understanding.

From the data shown in Fig. 2C, it appears that a pi pulse which inverts the population between $|11\rangle$ and $|22\rangle$ state takes about 300 ns. This is relatively slow comparing to the state-of-art in superconducting qubits, where such a pulse for a transmon qubit is typically 100 ns or less (see, for example, Krinner et al., Nature 2022). Therefore, it is desirable to reduce the pulse duration. Furthermore, this appears to be critical for improving the gate fidelity as the authors claimed that decoherence is the major source for gate infidelity.

However, it is not clear to me how much shorter this pulse could be reduced to given that the pulse ramping time is 100 ns which is required by the adiabaticity condition. It will be helpful to if the authors can elaborate on their plan to reduce the gate duration.

Response:

We appreciate the opportunity to further explain the physical mechanism behind the interaction and clarify the reported results.

The adiabaticity condition can be understood as arising from the need to transform the laboratory frame to the dressed frame. Specifically, consider the Hamiltonian of a single, off-resonantly driven qubit in the rotating frame of the drive: $H(t) = \Delta_q \sigma^z + \Omega_R(t) \sigma^x$. The drive has an envelope $\Omega_R(t)$. Due to the quantum adiabatic theorem, if Ω_R is slowly ramped up or down, the eigenstates of σ^z will evolve into the instantaneous eigenstates of $H(t)$. In practice, it is important to change $H(t)$ sufficiently slowly. Failure to do so leads to errors which can be understood as Landau-Zener transitions. The relevant parameter in such a transition is the rate at which the spectral gap of H is changing, which is, to low order in Ω_R/Δ_q , $\sim \Omega_R \dot{\Omega}_R/\Delta_q$. This tells us that at fixed Δ_q and Ω_R , we must decrease $\dot{\Omega}$ to fulfill adiabaticity as much as possible. As the reviewer correctly points out, the ramping rate can only be slowed down to the point that coherent error no longer dominates.

The data in Fig. 2c shows that a swap angle is achieved for a gate time of about 240 ns, which includes the 200 ns of ramping up and down. We would like to clarify the following:

- The 100 ns ramping is not the shortest ramping time possible for this specific $|11\rangle - |22\rangle$ two-photon experiment. In this demonstration, we chose a cosine ramp pulse with $\tau_r = 100$ ns so that all the reported data in Fig. 2d and Fig. 2e at various driving amplitudes could be obtained in the adiabatic regime, for consistency.
- The cosine ramp can be further optimized, for instance, by using Derivative Removal by Adiabatic Gate (DRAG), or by using uneven ramping. The pulse tuning depends on the specific ratio between the drive amplitudes, which subsequently determines the drive frequency. In order to focus on the dynamics of the interaction rather than the engineering and optimization of the pulses, we choose to work with a simple cosine ramp in this work. In a follow-up project, we will perform systematic measurements to understand the spectral property of the pulses and report relevant effects on the adiabaticity.
- To further reduce the ramping time, we can operate in the *diabatic* regime. We are currently investigating the generalization of this framework, but we can refer to some specific examples: (i) the geometric phase accumulations reported

in *Phys. Rev. X* 11, 021026 (2021), and (ii) the alignment of the oscillations suggested in *Nat. Phys.* 20 (2024).

- While it is true that our works have established a bound on the microwave-activated gate time due to adiabaticity, the actual figure of merit is the ratio between the interaction time and coherence time. While flux-tunable devices may have shorter interaction time, the coherence times are fundamentally limited by flux noise during operation. On the other hand, fixed-frequency devices generally have much longer coherence times. We have emphasized this point in the Discussion section. For example, planar fluxonium devices biased at half-flux quantum have been recently reported to have $T_1, T_2 \sim 1$ ms in [arXiv:2405.05481](https://arxiv.org/abs/2405.05481) (2024). While the experimental device we used in this work has relatively short coherence times, we envision that future devices with coherence times >500 μ s will exhibit high fidelities if the gate times are maintained at a range of 50-300 ns. We also note that the flip-flop dynamic appears to be coherent up to 1 μ s in both Fig. 2c and Fig. S6c, signifying the robustness of the protocol.
- The interaction is inherently limited by the coupling strength g_{kl} between the qudit levels. For static coupling case, this coupling is typically limited to a few MHz to keep the static cross-Kerr interactions sufficiently small. We believe that future devices integrating tunable couplers can dramatically enhance the effective coupling strength g_{kl} between the relevant levels, thereby increasing the interaction rate for a fixed set of driving amplitudes, which subsequently reduces the required ramping time. For example, we are currently investigating several strategies with positive outcomes and will report the results soon. Therefore, we envision that increasing the coupling during the interaction using fast tunable couplers will decrease the total gate times significantly.

To clarify these conceptual ideas, we have revised the manuscript as follows

- On page 3, we added the reason for choosing the stated ramping time, and how this can be reduced in the future.
- In Supplementary Material Note 2, we added a subsection on adiabaticity to clarify what it means and what are the requirements to satisfy this condition

Comment 3

In Fig. 2E, why the data shows relatively large deviation from the analytical mode for interaction rate between $|22\rangle$ and $|33\rangle$ state? It seems that the data for drive strength between 30 and 50 MHz are following a different scaling comparing to the rest.

Response:

We thank the Reviewer for pointing this out. We speculate that this is due to increased spurious effects due to spectral crowding in the higher transmon manifold which can lead to unexpected hybridization at certain driving strengths, but we do not fully understand this deviation at the moment. We have revised the text to point out this discrepancy. However, we prefer not discussing speculations.

Comment 4

In the section “Distribution of qudit entanglement”, it is shown that the unpurified quantum state fidelity of Bell-4 state (53.0%) is significantly lower than that of the Bell-3 state (75.7%). I would like the authors to provide a more detailed explanation for this dramatic reduction in state fidelity.

Response:

We thank the Reviewer for pointing out this detail. We should first clarify that in this section, we exclusively discuss experiments with Bell₃ states, not Bell₄.

The distribution is implemented by first creating a Bell state between Q_1 - Q_2 , then performing a sequence of swapping to *move* the entanglement across the array. The lower unmodified fidelity of the Q_1 - Q_4 pair compared to Q_1 - Q_3 can be attributed to the longer corresponding sequence. Considering that the Q_1 - Q_2 Bell₃ unpurified state fidelity is 97.7%, and that the Q_1 - Q_3 Bell₃ unpurified state fidelity is reduced to 75.7% (a gap of 22%), the reduction of Q_1 - Q_4 Bell₃ state fidelity to 53% is actually quite consistent.

We have also added the details on this sequence in Supplementary Material Note 8.

Comment 5

The authors attribute the infidelity of the purified quantum states to SPAM errors occurred in the tomography sequence. However, there is no discussion on SPAM errors in these experiments. I would like to ask the authors to provide more details on this topic to support their assessment.

Response:

We thank the Reviewer for pointing this out. The statement that the errors remained after purification is due to state-preparation-and-measurement is not entirely correct, and we regret making this mistake.

In general, the errors observed from state tomography can be due to (i) heralding to prepare the qudits in $|0\rangle^{\otimes n}$, (ii) unitary errors in executing the two-photon interactions used in preparing the intended states, (iii) local rotation errors in preparing the states, (iv) projection errors during tomography, and (v) readout errors. Purification effectively cancels the effects from decoherence, so the remaining errors can be from unitary errors and leakage in both preparing the states and tomography projections. We note that the crosstalk errors for $d = 4$ was not effectively mitigated at the time of the Schrödinger cat experiment. Thus, the fidelity of $(|000\rangle + |333\rangle)/\sqrt{2}$ is noticeably lower than that of $(|00\rangle + |33\rangle)/\sqrt{2}$ due to imperfect simultaneous local rotations of the three qudits, especially the $|2\rangle - |3\rangle$ gates. We are currently developing a dynamical decoupling technique for qudits, which we expect to improve the local control fidelity in the future. This work will be reported in another manuscript soon.

Since the SPAM errors include both decoherence and non-decoherence types, we have now changed our description. We have corrected the statement and added relevant details in Supplementary Material Note 6 regarding local control errors to reflect this understanding.

Responses to Reviewer #3

Comment 1

I find the manuscript fairly well written and pedagogical although some parts, in particular some of the terminology the authors use, require further clarification. The experiment itself is a true tour-de-force and the data is well analyzed and supported by an extensive supplementary material. While the conceptual novelty of the $|k,l\rangle \leftrightarrow |k+1,l+1\rangle$ transition is rather modest, the authors use it to generate highly entangled qudit states with very good fidelity. I believe that this work could be of interest to the broad readership of Nature Communications, but, before making a recommendation, I would like the authors to address the following points.

Response:

We thank the Reviewer for the positive feedback and encouraging comments. We greatly appreciate the suggestions provided, which have significantly enhanced the clarity and depth of our manuscript. We hope our revisions meet your expectations and look forward to the possibility of our paper being published soon.

Comment 2

The authors convincingly outline in the supplementary materials that this approach does not rely on the anharmonic spectrum of a transmon circuit but instead focuses on transitions through virtual states and is therefore more general. However, their experiments are of course performed with transmon circuits and so it is natural to compare to previous experiments in the field. In particular, the monochromatically driven $|0\rangle \leftrightarrow |1\rangle$ transition in ref 34 seems very similar to the transition shown here. Could the authors please comment in-depth on the similarities and differences between these two transitions?

Response:

We thank the Reviewer for pointing out the connection to Ref. 34. The two-photon transition in Ref. 34 is facilitated by a hybridization involving the $|02\rangle$ state. This was deliberate and achieved by designing the sample such that the frequency of the $|11\rangle$ state is very close (detuned by ~ 9 MHz) to that of the $|02\rangle$ state (see Table I of Ref. 34). The consequence of this hybridization is that the oscillation rate Ω_B in Ref. 34 acquires corrections compared to the pure qubit limit (see Eq. (3) in Ref. 34). In contrast, our scheme does not rely on this hybridization to achieve a fast two-photon oscillation. Further, our analytical approach better describes the physics of the two-photon oscillation at high drive power.

We have amended the discussion after Eq. S12 in Supplementary Material Note 2 to clarify this.

Comment 3

Following up on the previous question: as far as I can see, these gate processes could be framed in terms of a four-wave mixing process enabled by the transmon nonlinearity. In that case the resonance condition would simply be that the two transition frequencies ($k,l \leftrightarrow k+1,l$ and $k,l \leftrightarrow k,l+1$) need to be equal to the sum of two drive frequencies ($\omega_1 + \omega_2$). One could then see the process the authors demonstrate as the special case where $\omega_d = \omega_1 = \omega_2$. Maybe the authors could explain if this is indeed the case? Does their system have capabilities that go beyond this four-wave mixing description in coupled transmons?

Response:

We thank the Reviewer for suggesting this interesting connection to another framework. We believe that the four-wave mixing condition described above is not equivalent to the dynamics discussed in this work. First, the nonlinearity of the transmon is unimportant, except insofar as it allows us to speak meaningfully about a qubit. The relevant nonlinearity in the qubit problem is that two otherwise linear modes are coupled. Second, when the qubits are driven symmetrically, the resonance condition is given by $\omega_d = (\omega_{q,1} + \omega_{q,2})/2$, the average qubit frequency. Finally, when the qubits are driven asymmetrically, the resonant frequency ω_d can be pushed up or down depending on whether the lower or higher qubit frequency is driven more strongly, as given in Eq. (S11) in Supplementary Note 2. Together with our emphasis toward the end of the main section entitled “Traversing the dual bosonic ladder”, we believe that the manuscript provides sufficient information for the readers to appreciate the novelty of our two-photon dynamics framework.

Comment 4

The authors state that “the optimal drive frequency can be tuned in situ by adjusting the drive amplitudes and their relative ratio.” As far as I can see, the tuning of the optimal drive frequency is not shown experimentally in this work. If such data exists it would be an interesting addition to the SI. Otherwise, the statement should maybe be amended to clarify that it refers to a theoretical prediction.

Response:

We thank the Reviewer for raising this concern. The claim in the Discussion may indeed seems unsubstantiated.

To best present our findings in a compact and coherent fashion, we showcase in Fig. 2d the change of the optimal driving frequency with respect to different driving amplitudes when the two drives are not applied evenly to the qudits. Our model described in the SI takes this into account and shows excellent agreement with the data. In the SI, we then use this model to further display numerically how the optimal frequency can change as we tune the parameters of the drives. We have modified the text in Discussion and Supplementary Material Note 2 to reflect this. Finally, we have modified the statement to be more modest.

Comment 5

“native gate fidelities of $F_g = 0.99936(3)$, $0.99909(4)$, $0.9978(2)$ for $d = 2, 3, 4$,” Are the fidelities extracted from the fit to the data in figure 1b? If so, please mention this and add a reference.

Response:

We extracted the Clifford fidelities by fitting to the decay curves corresponding to the randomized benchmarking results (Fig. 1b, bottom panel). The average native gate fidelities can then be found by divided the Clifford errors by the number of gates per Clifford. We have added the detail in Supplementary Material Note 1 and refer to this in the main text.

Comment 6

Are the lines in Fig1b and 2c fits or a guide for the eye?

Response:

The oscillations are fitted using a cosine function, and the decays are fitted using an exponential functions. The fit are then shown as solid lines. We have modified the captions to reflect this.

Comment 7

“(Bottom) Preparation and measurement confusion matrix of qudit states, which reflects the readout fidelities.” Could the authors please give a definition in SI or cite a reference.

Response:

We have added the definition in Supplementary Material Note 1.

Comment 8

“While transitions involving single-photon hopping are permitted, the direct two-photon hopping process $|k, 1\rangle \rightarrow |k + 1, 1 + 1\rangle$ is prohibited due to parity.” This could be formulated more clearly. Presumably the authors mean the lack of parity conservation under a coherent drive?

Response:

We thank the Reviewer for pointing out a possibly confusing statement. We have modified the text as suggested, “due to the lack of parity conservation under a coherent drive”.

Comment 9

Fig2: was $\delta\omega_d$ defined in the text or figure caption? What is meant by symmetric point? The data looks asymmetric.

Response:

We have modified the caption as follows. Instead of “symmetric point”, we now refer to the $\delta\omega_d = 0$ point as optimal frequency point where the flip-flop dynamics is the most coherent.

Comment 10

Fig2 caption and other occurrences throughout the text and supplementary materials: “The y-axis is normalized with respect to omega median,” In what sense is this a normalization?

Response:

We have modified this as “with respect to ω_{avg} ”.

Comment 11

“median qudit frequency” What is meant by median in this context? Do the authors perhaps mean average?

Response:

We thank the Reviewer for raising this concern. Median and average have the same meaning here, as we have defined in the caption. It is the sum/2 of the two relevant single-photon frequencies, $(\omega_{k,l+1} + \omega_{k+1,l})/2 = (\omega_{k+1,l+1} - \omega_{k,l})/2$.

To clarify, we have now changed the notation to “average” and added the definition $\omega_{\text{avg}} = (\omega_{k,l+1} + \omega_{k+1,l})/2$

Comment 12

Figure 2d: Why is this frequency shift positive? If ω_{median} is the undriven average qubit frequency and the anharmonicity is negative, then the a.c. Stark shift will generally also be negative. This means that the optimal drive frequency given by the new (smaller) average qubit frequency should be smaller than ω_{median} . Is this positive frequency shift of the resonance condition due to the fact that Q2 is driven in the straddling regime (between the frequencies of the 0 \leftrightarrow 1 and 1 \leftrightarrow 2 transitions)? What are the expected frequency shifts of the individual 0-1 transitions of the two qubits under this drive?

Response:

The origin of the shift here is primarily due to the asymmetric drive, not due to the anharmonicity of the qubit. As demonstrated in Suppl. Mat. Note 2, even in the pure qubit limit, the optimal drive frequency ω_d will depend on the drive amplitude Ω when the two qubits are driven with unequal power. The sign of the shift $\omega_d - (\omega_{q,1} + \omega_{q,2})/2$ is positive (negative) when the higher (lower) frequency qubit is driven more strongly than the lower (higher) frequency qubit. We have added a sentence to emphasize this point in Suppl. Mat. Note 2.

Comment 13

“which can be accounted for by our analytical model (Fig. 2d).” Please specify which equation is used to get this curve. What are the fit parameters? Is the microwave crosstalk indeed accounted for by that model?

Response:

We thank the Reviewer for the request for clarification. Since the full analytical model is included in the Supplementary Material, we revised the text to guide the readers to Suppl. Mat. Note 2. In addition, we have now explicitly mentioned which equations in Suppl. Mat. Note 2 are used in the fitting to indicate the fit parameters. The model indeed accounts for microwave crosstalk through the λ parameter, which results in an asymmetric drive on the two transmons and, consequently,

a significant shift in the resonance frequency ω_d . This is pointed out in the main text, and we have added a short remark to that effect in Suppl. Mat. Note 2 as well.

Comment 14

Same as above for Fig 2e :” Our analytical model again provides an excellent fit to the experimental data.”

Response:

We included the same information as for Fig. 2d.

Comment 15

“access to otherwise impossible operations” Please explain why these operations would otherwise be impossible.

Response:

We have modified the sentence to be more moderate. The operations would be difficult and resource-extensive to implement, but not impossible. We also narrow down to Toffoli and CnZ gates. We can only speak for these implementations, which are already discussed more thoroughly in Refs. 23 and 24.

Comment 16

“is FCCCZ = 92(1)%, on-par with the best results previously reported in superconducting qubits [24] and trapped-ions [26].” It would be good to also refer to the lower fidelity estimate given in the SI (88%) by a different method.

Response:

We have added this detail.

Comment 17

“for novel fault-tolerance schemes” Incomplete sentence?

Response:

We thank the Reviewer for pointing out this ambiguous wording. We have polished the language.

Comment 18

All figures that include bar graphs: it seems like only the bars corresponding to the density matrix elements that would be non-zero for an ideal state are coloured blue. This is somewhat visually misleading. It would be better to find a different color scheme and to include wireframe boxes depicting the ideal values

Response:

We thank the Reviewer for the opportunity to discuss our choice of visual displays of the data. We were inspired by Fig. 2 in J. Wang et al., Science 360 (2018) and Fig. 3 in Y. Chi et al., Nat. Commun. (2022), which show the data using three-dimensional bar plots. With high fidelities (as reported along side the plots), the important terms are sufficiently emphasized. We believe that reporting the fidelities in this context provides the most important information to the general readers. We further note that we provide the reported data to the public in matrix format, accompanied by the ideal matrices. Experts interested in perusing the obtained data can access them liberally.

However, the color setting of Fig. 2 in J. Wang et al., Science 360 (2018) (or Fig. 3 in Y. Chi et al., Nat. Commun. (2022)) makes it difficult for the readers to see the important terms in the plots. Our improved color scheme works as follows. We assign a threshold for the matrix entries. Below this threshold, the bars are assigned the color gray. Above this, they are assigned the color blue. Since only the blue entries contribute to the fidelities (see A. Cervera-Lierta et al., Phys. Rev. Applied

17 (2022) for a more systematic discussion), this color scheme helps to emphasize the important terms and allow the readers to see them more clearly. In combination, we believe that our approach allows the data to be visualized in a clear, aesthetically appealing, and visually efficient fashion without compromising the technical details, which are summarized by the reported fidelities and provided to the more specialized readers via accessible datasets.

We have added a description in Supplementary Material Note 5 to address the construction of the tomography plots.

Comment 19

SI Figure S5: “(b) Two-photon swap rate Ω_{2p} as a function of the drive amplitude and qubit detuning Δ_{12} . Even when the energy levels $|k+1, 1\rangle$ and $|k, 1+1\rangle$ are far-detuned, it is possible to activate the two-photon swap, albeit requiring more drive power.” For what drive frequency is this simulated?

Response:

The numerics are performed in the symmetrically driven case, as indicated by the text “ $\lambda = 1.0$ ” in the top right corner of Fig. S5b, which implies that ω_d is the average of the two single photon transition frequencies, following Eq. (S11). Since it is in the symmetrically driven limit, it is equivalent to $4J_I$ as defined in Eq. (S13). As such, the absolute value of the drive frequency does not enter. We have updated the caption as well as the final paragraph of subsection “Theory of coupled two-level systems” in Suppl. Mat. Note 2 to emphasize this point.

Comment 20

SI: “and compute the energy difference $\omega^{(i)}_q$ between the states $|0\rangle$ and $|1\rangle$.” Is this what is meant by exact diagonalization (numerical ED) in Fig. S5? If so, please mention that here.

Response:

We thank the Reviewer for pointing out the ambiguity of the legend in Fig. S5c. We have amended the text to clarify that the method “numerical ED” refers to exact diagonalization of the Duffing oscillator Hamiltonian in the rotating frame (Eq. (S15)).

Comment 21

The visual representations on Fig.S1 give a good qualitative overview of the system parameters, but they are not quantitative enough. Could the authors please also add a table including frequencies, anharmonicities, coupling strengths between qubits as well as qubits and resonators, linewidths of resonators. Please also add average T1,T2,T2E.

Response:

Yes, certainly. We added a table listing the experimentally obtained parameters.

Comment 22

A scale bar is missing in figure S1a

Response:

We thank the Reviewer for pointing this out. We have added the scale bar.

REVIEWERS' COMMENTS

Reviewer #2 (Remarks to the Author):

I thank the authors for detailed response to my comments in the previous round of review. The revisions in both the main text and the supplemental information are satisfying. The clarity and accuracy of the manuscript are improved. Therefore, I recommend accepting the current manuscript for publication in this journal.

Reviewer #3 (Remarks to the Author):

I would like to thank the authors for meaningfully addressing my comments and questions. I can recommend the manuscript for publication in its present form. I have two minor comments/questions that the authors could still address:

Equation S(11): Missing subscript on Omega?

Discussion of Fig.2d and Fig.2e: I could not find the value of the fixed strength of the drive applied to the second qubit in the manuscript. It would be informative to give that value as well.

Finally, I would like to point out one observation relating to the author's response to comment 3:

"Second, when the qubits are driven symmetrically, the resonance condition is given by $\omega_d = (\omega_{q,1} + \omega_{q,2})/2$, the average qubit frequency" : At least for the transition $0,0 \leftrightarrow 1,1$ this is indeed the frequency condition that would be required by a four-wave mixing process.

Energy conservation dictates that the two drives I mentioned in my comment need to fulfil: $\omega_{d,1} + \omega_{d,2} = \omega_{q,1} + \omega_{q,2}$. If one sets $\omega_d = \omega_{d,1} = \omega_{d,2}$ the resonance condition for both drives becomes $2\omega_d = (\omega_{q,1} + \omega_{q,2})$ as the authors suggest. However, I think that the correction coming from asymmetric driving is indeed not accounted for by a simple four-wave mixing model.

Dear Reviewers:

Thank you for your efforts in reviewing the revised manuscript "Empowering a qudit-based quantum processor by traversing the dual bosonic ladder". We are glad to receive positive feedback from Reviewers #2 and #3. We have further revised the manuscript according to Reviewer #3's suggestions and editorial requests. Please see below the summary of our modifications.

Modifications according to Reviewer #3's comments

- Supplementary Note 2: we defined the notations regarding driving amplitude Ω .
- Supplementary Note 2: we provided information on the fitted λ values.

Modifications according to editorial guidelines

- The main text is now organized following this order:
 - Title
 - Author list
 - Affiliations
 - Abstract
 - Introduction
 - Results with separate subsections
 - Discussion
 - Data availability
 - References
 - Acknowledgments
 - Author contributions
 - Competing interest statement
 - Figure captions
- Changed "Supplementary Materials" to "Supplementary Information". The Supplementary Information file is provided as a separate PDF file with a full reference list.
- All figures in the Supplementary Information are labeled with "Supplementary Figure #".
- All tables in the Supplementary Information are labeled with "Supplementary Table #".
- All equations in the Supplementary Information are labeled as "#", not "S#".
- All references to Supplementary Information sections are changed to "Supplementary Note #".
- All references to figures in Supplementary Information are changed to "Supplementary Figure #".
- All references to equations in Supplementary Information are changed to "Supplementary Equation #".
- All references to tables in Supplementary Information are changed to "Supplementary Table #".

Response to Reviewer #2

Comment 1

I thank the authors for detailed response to my comments in the previous round of review. The revisions in both the main text and the supplemental information are satisfying. The clarity and accuracy of the manuscript are improved. Therefore, I recommend accepting the current manuscript for publication in this journal.

Response:

We thank the Reviewer for the encouraging remark and the recommendation for publication.

Responses to Reviewer #3

Comment 1

I would like to thank the authors for meaningfully addressing my comments and questions. I can recommend the manuscript for publication in its present form.

Response:

We thank the Reviewer for going through our modifications and the positive feedback.

Comment 2

Equation S(11): Missing subscript on Omega?

Response:

We thank the Reviewer for suggesting this. We have added the notations to indicate that $\Omega \equiv \Omega_2$ is the driving amplitude on one qubit, with λ serves to indicate the driving strength $\Omega_1 = \lambda\Omega_2 = \lambda\Omega$ to the other qubit.

Comment 3

Discussion of Fig. 2d and Fig. 2e: I could not find the value of the fixed strength of the drive applied to the second qubit in the manuscript. It would be informative to give that value as well.

Response:

In the discussion on fitting the data presented in Fig. 2, allocated inside Supplementary Information Note 2, we have added the detail regarding the drive ratio λ which relates the drive amplitudes on Q_1 and Q_2 .

Comment 4

Finally, I would like to point out one observation relating to the author's response to comment 3: Second, when the qubits are driven symmetrically, the resonance condition is given by $\omega_d = (\omega_{q,1} + \omega_{q,2})/2$, the average qubit frequency": At least for the transition $0,0 \leftrightarrow 1,1$ this is indeed the frequency condition that would be required by a four-wave mixing process. Energy conservation dictates that the two drives I mentioned in my comment need to fulfill: $\omega_{d,1} + \omega_{d,2} = \omega_{q,1} + \omega_{q,2}$. If one sets $\omega_d = \omega_{d,1} = \omega_{d,2}$ the resonance condition for both drives becomes $2\omega_d = (\omega_{q,1} + \omega_{q,2})$ as the authors suggest. However, I think that the correction coming from asymmetric driving is indeed not accounted for by a simple four-wave mixing model.

Response:

We thank the Reviewer for emphasizing this observation. From this perspective, the two-photon process can be viewed as a multi-wave mixing process: two energy quanta at different frequencies are converted to two other energy quanta at different frequencies. Indeed, in the current context, we are concerned that a *simple analogy* to four-wave mixing may mislead the readers.